# Determination, Modeling and Evaluation of Kinetics of ^223^Ra Sorption on Hydroxyapatite and Titanium Dioxide Nanoparticles

**DOI:** 10.3390/ma13081915

**Published:** 2020-04-19

**Authors:** Petra Suchánková, Ekaterina Kukleva, Karel Štamberg, Pavel Nykl, Michal Sakmár, Martin Vlk, Ján Kozempel

**Affiliations:** Faculty of Nuclear Sciences and Physical Engineering, Department of Nuclear Chemistry, Czech Technical University in Prague, Břehová 7, 11519 Prague 1, Czech Republic; petra.suchankova@fjfi.cvut.cz (P.S.); ekaterina.kukleva@fjfi.cvut.cz (E.K.); kstamberg@volny.cz (K.Š.); nyklpavel1@gmail.com (P.N.); michal.sakmar@fjfi.cvut.cz (M.S.); martin.vlk@fjfi.cvut.cz (M.V.)

**Keywords:** hydroxyapatite, titanium dioxide, radium-223, nanoparticles, ion sorption kinetics, kinetic models

## Abstract

Sorption kinetics of radium on hydroxyapatite and titanium dioxide nanomaterials were studied. The main aim of the current study was to determine the rate-controlling process and the corresponding kinetic model, due to the application of studied nanomaterials as α-emitters’ carriers, and to assess the sorption properties of both materials from the radiopharmaceutical point of view by time regulated sorption experiments on the nanoparticles. Radium-223 was investigated as radionuclide used in targeted alpha particle therapy as an in vivo generator. It was found that the controlling process of the ^223^Ra sorption kinetics was the diffusion in a reacted layer. Therefore, parameters like particle size, their specific surface area, contact time and temperature played important role. Moreover, the composition of liquid phase, such as pH, the concentration of ^223^Ra, ionic strength, the presence of complexation ligands, etc., had to be considered. Experiments were conducted under free air conditions and at pH 8 for hydroxyapatite and pH 6 for titanium dioxide in Britton–Robinson buffer. Initial ^223^Ra concentration was in the range from 10^−11^ to 10^−12^ mol/L. It was found that sorption kinetics was very fast (more than 90% in the first hour) in the case of both nanomaterials, so they can be directly used for efficient radium sorption.

## 1. Introduction

Both studied materials, hydroxyapatite (Hap) and titanium dioxide are situated among natural sorbents. Their properties were studied in the case of inorganic and organic contaminants’ separation from different types of waters [1,2,3,4,5]. Recently, the attention was devoted to nanoparticles (NPs) because of their large specific surface area that leads to relatively fast either sorption or desorption kinetics and high sorption capacity. Especially, the pH dependent sorption properties, based on the protonation or deprotonation of surface sites [1,6,7,8,9], enable the sorption of cationic or anionic species on studied nanoparticles by ion exchange and surface-complexation mechanism simultaneously [6,8,9,10,11].

Another advantage of both materials is their biocompatibility. Hydroxyapatite—(Ca_10_(PO_4_)_6_(OH)_2_)—is the natural component of bones and its synthetic analogue is widely used as artificial bones, adsorbent for protein chromatography, additive in toothpastes, etc. [2]. Several studies focusing on adsorption of various substances onto hydroxyapatite, such as icariin [12], organic dyes [13], Pb^2+^ [2,3,14], Cd^2+^ [2,15], etc., were already published. They were dealing with contaminant retention and purification. It was found that the retention kinetics of the Hap could be quite fast and strongly depending on the substance. For substances, which do not react with Hap—i.e., their retention mechanism has physical character—it took approximately 15 min [12,13]. In the case of cations, which could interact with the Hap structure, required time for the reaction was in the range from 15 min to 1.5 h for different ions. Sorption capacity also belongs to important parameters. It was found that sorption capacity of Hap was up to 500 mg/g for Pb [3,14]. Due to those facts, Hap can be used not only as a sorbent for contamination removal, but also as a carrier for different radionuclides [2,3,14,15].

Titanium dioxide is widely used in food technology as a pigment known as E171, moreover, TiO_2_ is used as a filling material in drugs or cosmetics. However, sorption properties of TiO_2_, which were studied in this paper, were being investigated for a very long time. For example, ^68^Ge/^68^Ga generators used in nuclear medicine are packed with titania [16]. Another way to use TiO_2_ as a sorbent was the treatment technique for pollutant removing from the environment. Several studies were involved in the research of suitable conditions for removing organic contaminants, such as dyes [4], dioxane, N-nitrosodimethylamine (NDMA), tris-2-chloroethyl phosphate (TCEP) and estradiol [17], oxalic acid [18], and inorganic ions, such as Zn^2+^ [5], As^3+^ and As^5+^ [19,20], Fe^2+^ [21], etc. mainly from water. According to the above-mentioned studies, sorption kinetics of TiO_2_ were rather fast, lasted from several minutes to few hours, and was depended on the type of contaminant. Sorption capacity of TiO_2_ was reported to be about 5 mg/g for the most ions [22].

The radioisotope ^223^Ra (T_1/2_ = 11.4 d) was chosen for the study due to its beneficial properties as an in vivo generator for targeted alpha therapy. Alpha emitting radionuclides are able to damage cells including DNA strand, and in the case of correct radionuclide-carrier combination could be very powerful in cancer treatment [23]. Radium-223 chloride (Xofigo^®^) is already used for palliative treatment of bone metastases of prostate cancer [24,25] due to its chemical behavior similar to calcium. Unfortunately, the application of Xofigo^®^ had to be restricted due to found side effects during ongoing clinical studies. Nowadays, Xofigo^®^ can be used only in several specific indications. This recommendation went from Pharmacovigilance Risk Assessment Committee (European Medicines Agency) [26]. Due to these recommendations and for the possibility of other organs treatment, it is necessary to create an appropriate carrier, which will immobilize the radionuclide and its decay products and drive into the tissue of interest. During the alpha decay a lot of energy is released and about 95% is carried by alpha particle (in the case of ^223^Ra it is approximately 5 MeV), and the rest, 5%, is carried by the new nucleus. Therefore, the bond between radionuclide and organic molecule is destroyed, because the nucleus energy is much higher than the energy of chemical bond [23]. For this and other reasons, e.g., the radiation stability, it is better to choose inorganic materials as carriers. So, in order to immobilize ^223^Ra and its daughter radionuclides, hydroxyapatite (*n*HAp) and titanium dioxide nanoparticles (*n*TiO_2_) were chosen.

The subject and goal of this study can be conceived as follows: (a) to summarize sorption kinetic models derived for two phase systems, (b) to evaluate the time dependent concentrations of ^223^Ra^(II)^ with the aim of identifying the rate-controlling sorption process and to find the corresponding kinetic model, and (c) to assess the influence of the basic reaction parameters. Surface protolytic properties [6], dependence of sorption on pH [10], and sorbed species of ^223^Ra [10] were already studied and described in detail. Current paper continues the study on *n*HAp and *n*TiO_2_ as perspective materials for nuclear medicine as a radionuclide vehicle.

## 2. Materials and Methods 

All chemicals were of analytical grade purchased from Merck (Darmstadt, Germany) and were used without further purification: tetrabutyl *ortho*-titanate (TBOT), prop an-2-ol (IPA), sodium hydroxide, phosphoric acid, boric acid, acetic acid, sodium nitrate, ammonium hydroxide solution (28%), calcium nitrate tetrahydrate, and ammonium hydrogen phosphate. Demineralized water of 18 MΩ/cm was obtained from Millipore, (Burlington, MA, USA) water purification system.

Radioactivity measurement was performed on CRC-55tW (CAPINTEC, Ramsey, NJ, USA) calibrated detector. During the experiments, all samples were mixed with curved glass stick fixed in IKA RW 11 stirrer (Staufen, Germany), centrifugation was performed on Centrifuge MPW-360 (MPW, Warsaw, Poland).

### 2.1. Preparation of ^223^Ra Stock Solution

Radium-223 generator (^227^Ac/^227Th^/^223^Ra) was prepared in our laboratory according to Guseva et al. [27]. The elution was performed by 0.7 M nitric acid in 80% methanol solution from column (0.5 g of Dowex-1 × 8 and ^227^Ac in equilibrium with its decay products) for the separation of ^223^Ra stock solution from ^227^Ac and ^227Th^. The gained ^223^Ra in the form of ^223^Ra(NO_3_)_2_ was dried and reconstituted with saline. No breakthrough of parent radionuclides was observed in γ-spectrum of the eluate.

### 2.2. Britton–Robinson Buffer Solution

The necessary Britton–Robinson buffer (BRB) was prepared from two stock solutions, which were mixed in an appropriate ratio. The first one was 0.2 M sodium hydroxide and the second one was the mixture of 0.04 M phosphoric acid, 0.04 M boric acid, and 0.04 M acetic acids.

### 2.3. Sorbent Material Preparation

Hydroxyapatite nanoparticles were prepared by using equal volumes of 1.2 M Ca(NO_3_)_2_ and 0.7 M (NH_4_)_2_HPO_4_ in order to obtain the Ca:P = 1.67 ratio. At first, the calcium salt solution was added to 0.5 L of demineralized water. The pH of the mixture was set to 11 and maintained during the reaction. Then (NH_4_)_2_HPO_4_ was added dropwise during stirring. After overnight stirring, gained NPs were washed (demineralized water–three times) and dried under vacuum. 

Mixture of TBOT and IPA (1:4) was used for titanium dioxide nanoparticles preparation. The mixture was added dropwise into excess of demineralized water under ultrasonication. The whole mixture was in ultrasonic generator for 30 min. Gained NPs were washed (demineralized water–three times, IPA–once) and then dried under vacuum. Detailed description of the materials preparation and characterization was published by Kukleva et al. [6] earlier.

### 2.4. Determination of Kinetic Dependences

The pH of following kinetic experiments was set to eight for *n*HAp and to six for *n*TiO_2_ according to the study of ^223^Ra uptake mechanism by NPs as a function of pH described by Suchánková et al. [10] and future planned use of these materials in medicine in vivo. Prepared sorbents were dispersed in 25 mL of BRB, so the concentration of NPs was 1 g/L. Then ^223^Ra(NO_3_)_2_ solution of similar pH was added, so average activity per sample was 200 kBq. The suspension was continuously stirred during the whole experiment (24 h). Small aliquots of suspension (0.5 mL) were taken from the sample in 1, 2, 3, 4, 5, 6, 7, 8, 9, 10, 15, 20, 25, 30, 40, 50, and 60 min and then after 2, 3, 4, 5, 7, 10, and 24 h, so the solid to liquid phase ratio remained constant. The zero aliquot was taken immediately after ^223^Ra solution addition. The aliquots were centrifuged for 30 s at 3000 rpm in the case of *n*HAp and 40 s at 4000 rpm in the case of *n*TiO_2_, due to different stability of nanomaterial dispersion. After separation, nanoparticles were redispersed in 0.5 mL of water to ensure uniform counting geometry. The radiochemical yield (*Y %*) of adsorbed ^223^Ra was calculated based on activity measurements of separated supernatant (*A_s_*) and nanoparticles (*A_p_*) according to Equation (1):(1)$$Y\;\left(\%\right)=\frac{A_{s}}{A_{s}+A_{p}}\times 100\%$$

Based on ^223^Ra absolute activity the concentration in the solution was calculated using Equation (2):(2)$$c=\frac{A\times T_{1/2}}{\ln2\times V_{aq}\times N_{A}}\;\left[mol\cdot L^{-1}\right]$$
where *A* (Bq) is the absolute activity of the radionuclide, *T*_1/2_ (s) is the radionuclide half-life, *V_aq_* (L) is the volume of the solution, and *N_A_* is the Avogadro constant.

The experiments were repeated three times for each sorbent.

### 2.5. Kinetic Models for Two-Phase Systems

Kinetic models are summarized in Table 1 [28] and reflect the following different rate-controlling processes: mass transfer (DM), film diffusion (FD), diffusion in inert layer (ID), diffusion in reacted layer (RLD), chemical reaction (CR), and gel diffusion (GD). All of these models are given by first order differential equations.

### 2.6. Procedure of the Experimental Data Evaluation

The RLD-model was used to demonstrate calculations of the liquid–solid sorption kinetics in detail (Equations (8) and (9)). The systems were following ^223^Ra^(II)^–*n*HAp and ^223^Ra^(II)^–*n*TiO_2_. Sorption from the aqueous phase into the solid sorbent is described by the Equation (8), which was modified with balance and equilibrium equations (Equations (15) and (18)). The obtained equation can be used for direct evaluation of the experimental data. The value of the over-all mass transfer coefficient, *K_RLD_*, was sought based on the Newton–Raphson multidimensional non-linear regression method combined with the solution of the differential equation under given boundary conditions (Runge–Kutha method). Calculations were performed on in-house made code P60.fm of software product FAMULUS (Famulus Etc., Prague, Czech Republic) [28].

If the value of WSOS/DF (weighted sum of squares divided by degrees of freedom) was in the range from 0.1 to 20 [29], the evaluated model was taken as suitable. The calculations were based on the *χ*^2^—test according to Equation (19). Based on Equation (20) the value of WSOS/DF was obtained.
Χ^2^ = ∑{(*SSx*)_i_/(*s_q_*)_*i*_^2^}}; i = 1, 2, 3, …, *n_p_,*(19)
*WSOS*/*DF* = (χ^2^/*n_d_*); *n_d_* = *n_p_* − *n*,(20)
where (*SSx*)_*I*_ is the i-th square of the deviation of i-the experimental value from the corresponding calculated one, (*s_q_*)_*I*_ is the estimate of standard deviation (uncertainty) of the i-the experimental point, *n_p_* is the number of experimental points, *n_d_* is the number of degrees of freedom, and *n* is the number of model parameters sought during the regression procedure.

## 3. Results and Discussion

### 3.1. Parameters of nHAp and nTiO_2_

As mentioned above, the preparation and characterization of the sorbents was published earlier by Kukleva et al. [6] and the pH values were chosen in terms of ^223^Ra sorption mechanism results published by Suchánková et al. [10]. Characterization parameters of the sorbents are summarized in Table 2. The significant differences can be seen in the values of specific surface area determined by the Brunauer–Emmett–Teller (B.E.T.) method, crystallite size and equivalent diameter of NPs. Based on this data, higher kinetic rate should be expected in the case of *n*TiO_2_ than of *n*HAp. However, *n*HAp has higher concentration of surface edge sites, which can be interpreted as its potential higher sorption capacity for species present under pH ≥ 8–9 [6], but the radionuclide concentration was very low (Table 3), so this advantage has probably not played an important role.

### 3.2. Evaluation of Kinetic Dependence

The parameters used as input data, in addition to experimental time dependent concentrations (Figure 1a,b), are summarized in Table 3. Values of *K_d_* were obtained from equilibrium values of given kinetic experiments.

Primarily, each of the experimental kinetic dependence was fitted by all six models one by one (Table 1) and then evaluated based on the WSOS/DF value (Table 4). According to WSOS/DF values, only two models (ID and RLD) were found suitable for further modeling and description of the experimental results. However, RLD model better corresponds with the studied systems from the physical–chemical point of view, therefore it was chosen for further evaluation. Namely, the diffusion through reacted layer seems to be more realistic than through the inert one.

The experimental and calculated data are illustrated as time dependent percentage fraction of ^223^Ra sorbed on *n*HAp (Figure 1a) and on *n*TiO_2_ (Figure 1b) together with error bars. Due to the fast sorption kinetics, frequent sampling, very low radium concentration, and precautions applied for the work with radioactive materials, experimental errors, i.e., error bars, seem to be relatively high. However, all three repeated experiments for both materials were evaluated separately and the resulted mean values of: overall mass transfer coefficient, *K_RLD_*; diffusion coefficient, *D*, and sorption half-life, *t*_1/2_, are summarized with their correspond values of standard uncertainties, *σ_es_*, in Table 5 (the results also include the mean *K_d_* values). As for the sorption half-life, *t*__1/2__, it represents the time needed for a sorption to increase by half compared to its maximal (equilibrium) level in the given kinetic experiment. The diffusion coefficient in the solid phase, *D*, is determined using Equation (9) (mean radius of the solid phase particle, *R*, and its specific mass density, *h_s_*, can be found in Table 2). It is obvious that among the parameters mentioned above the most important, from the kinetic point of view, especially for NPs is mean radius, *R*, in addition to specific surface area (Table 2). Diffusion coefficient, *D*, depends on the temperature, type, and speciation of given component (in this case of ^223^Ra as RaHPO_4_, RaH_2_PO_4_^+^_,_ and Ra^2+^) [10,30]. Distribution coefficient, *K_d_*, is responsible for the value of driving force of the given transport process ((*q** − *q*) or (*c* – *c**) in Table 1). The larger *K_d_*, the larger driving force especially during first seconds or minutes and the larger positive contribution to the rate of kinetic process. This contribution to the increase in sorption rate is, as we believe, characterized by the half-life *t*_1/2_. If we compare the mean values of *t*_1/2_ ± *σ* (Table 5) with the mean values of *K_d_ ± σ* (Table 3) then we can conclude that the mean values have approximately comparable statistical significance and that their values correlate with each other. In our opinion, this indicates the positive impact of *K_d_* on the rate of the sorption process.

Comparing the mean values of *K_RLD_* (Table 5), at first glance it seems that the sorption on *n*HAp ought to be faster than on *n*TiO_2_. Of course, it does not hold true because the difference in *K_RLD_* values is in consequence of the reversible character of the sorption process studied. Especially in this case, it is the dependence of the sorption rate on the driving force given by the value *K_d_*, as mentioned above. The half-lives of the sorption reaction, *t*_1/2_ (Table 5), were considered as the quantification of this effect, the mean values of which were 0.75 ± 0.18 min and 0.51 ± 0.32 min for *n*HAp and *n*TiO_2_ respectively. Based not only on these values, but also on the values of *R*, *K_d_*, and specific surface area, the sorption kinetics on *n*TiO_2_ can be validated as faster than on *n*HAp.

## 4. Conclusions

The sorption kinetics of ^223^Ra to *n*HAp and *n*TiO_2_ were experimentally studied under free air conditions at 22 ± 1 °C. (a) There were summarized sorption kinetic models for the two phase system. The results were evaluated using six different kinetic models of a two-phase system, derived for six different rate control processes, based on first-order differential equations. (b) According to goodness-of-fit values, WSOS/DF, the model based on diffusion in the reacted layer (RLD) of the solid phase was chosen as the best for both types of sorbents studied. (c) Further, the influence of the basic reaction parameters was assessed. It was found that *n*TiO_2_ had higher value of *K_d_* compared to *n*HAp, which could be understood not only as higher sorption capacity for ^223^Ra under given reaction conditions, but also as higher time dependent driving force in the course of given kinetics, which contributes to a higher sorption rate on *n*TiO_2_. This was confirmed by the values of half-life reaction quantity, *t*_1/2_, *n*TiO_2_. This corresponded with the larger specific surface area of *n*TiO_2_ and its smaller particle diameter. Unfortunately, the values of overall kinetic coefficients, *K_RLD_*, on the basis of the first quick look, pointed to the opposite result. However, it is necessary to take into account the reversible nature of the sorption reaction, which implies that the parameter of the sorption kinetics is not only *K_LRD_*, but also *K_d_*, i.e., the driving force of the given reaction. In any case, the ^223^Ra sorption rate is sufficient and serves the purpose of *n*HAp and *n*TiO_2_ for medicinal and industrial applications.

## Figures and Tables

**Figure 1 materials-13-01915-f001:**
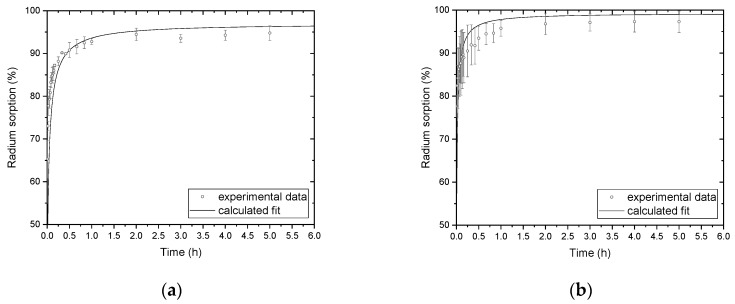
The kinetic of ^223^Ra sorption on nanoparticles (NPs) during the first hour: (**a**) *n*HAp (mean initial concentration c_0_ = 1.02 × 10^−11^ mol/L) and (**b**) *n*TiO_2_ (mean initial concentration c_0_ = 1.01 × 10^−11^ mol/L).

**Table 1 materials-13-01915-t001:** Kinetic models of sorption/extraction taking place in two-phase systems [28].

Control Process	Model Abbreviation	Differential Equation	Equation #
Mass transfer	DM	d*q*/d*t* = *K_DM_* × (*q** − *q*)	(3)
Film diffusion	FD	d*q*/d*t* = *K_FD_* × (*c* − *c**)	(4)
		*K_FD_* = 3 × *D*/(δ × *R* × *h_s_*)	(5)
Diffusion	ID	d*q*/d*t* = *K_ID_* × (*c* − *c**)/{[1 − (*q*/*q**)]^−1/3^ − 1}	(6)
in an inert layer		*K_ID_* = 3 × *D*/(*R*^2^ × *h_s_*)	(7)
Diffusion	RLD	d*q*/d*t* = *K_RLD_* × (*q** − *q*)/{[1 − (*q*/*q**)]^−1/3^ − 1}	(8)
in a reacted layer		*K_RLD_* = 3 × *D*/(*R^2^* × *h_s_*)	(9)
Chemical reaction	CR	d*q*/d*t* = *K_CR_* × *r_CR_* × [1 − (*q*/*q**)]^2/3^	(10)
in the reaction zone,		*K_CR_* = 3/(*R* × *h_s_*)	(11)
e.g., 1^st^ order reversible reaction		*R_CR_* = K*_CR_* × (*c* − *c**)	(12)
Gel diffusion	GD	d*q*/d*t* = *K_GD_* × [(*q** − *q*_0_)^2^ − (*q* − *q*_0_)^2^]/(*q* − *q*_0_)	(13)
		*K_GD_* = *D* × π^2^/(2 × *R*^2^)	(14)
Following balance equations hold:			
d*q*/d*t* = −*r* × d*c*/d*t*			(15)
if *c* is the integration variable: *q* = *r* × (*c*_0_ − *c*) + q_0_			(16)
if *q* is the integration variable: *c* = *c*_0_ − (*q* − *q*_0_)/*r*			(17)
Equilibrium equations used: *q** = *K*_d_ × *c*, and *c** = *q*/*K*_d_			(18)

*c* is a concentration of the component in the aqueous phase at time *t*; *c**—equilibrium concentration of the component in the aqueous phase corresponding to the concentration of the component in the sorbent at time *t*; *q*—concentration of the component in the sorbent at time *t*; *q**—equilibrium concentration of the component in the sorbent corresponding to the concentration of the component in the aqueous phase at time *t*; *q*_0_—starting concentration of the component in the sorbent; *t*—time; *r*—volume ratio of aqueous to solid phase; *D*—diffusion coefficient of the component, *K_DM_*; *K_FD_*, *K_ID_*, *K_RLD_*, *K_CR_*, and *K_GD_*—over-all kinetic coefficients*; k_CR_*—kinetic coefficient of the chemical reaction; *r_CR_*—rate of the chemical reaction; *R*—mean radius of the solid phase particle; *h_s_*—specific mass of the solid sorbent; and δ—thickness of the “liquid film” on the surface of the solid particle.

**Table 2 materials-13-01915-t002:** The basic properties of *n*HAp and *n*TiO_2_ [6,10].

Parameter	Unit	*n*HAp	*n*TiO_2_
Specific surface area	(m^2^·kg^−1^)	117 ± 8	330 ± 10
Crystallite size	(nm)	5.18	2.64
Equivalent diameter	(nm)	21.7 ± 6.9	5.3 ± 1.7
Surface edge sites	(mol·kg^−1^)	5.10 ± 1.20	0.20 ± 0.01
Surface layer sites	(mol·kg^−1^)	0.15 ± 0.01	0.67 ± 0.01
pH applicability	pH	5–10	2–10
Specific mass	(g·cm^−3^)	3.14–3.21	3.90–4.30

**Table 3 materials-13-01915-t003:** Starting aqueous phase concentrations of ^223^Ra, *c*_0_, and the values of parameters *K*_d_ and *r* in kinetic experiments of sorption on *n*HAp and *n*TiO_2_ (at *t* = 0).

Experiment No.	*n*HAp			*n*TiO_2_		
	*c* _0_	*K* _d_	*r*	*c* _0_	*K* _d_	*r*
	(mol·L_aq_^−1^)	(L_aq_∙kg^−1^)	(L_aq_∙kg^−1^)	(mol·L_aq_^−1^)	(L_aq_∙kg^−1^)	(L_aq_∙kg^−1^)
1	9.92 × 10^−12^	3.03 × 10^4^	1000	1.01 × 10^−11^	5.90 × 10^4^	1000
2	9.46 × 10^−12^	1.55 × 10^4^	1000	1.05 × 10^−11^	1.25 × 10^5^	1000
3	1.12 × 10^−11^	1.54 × 10^4^	1000	9.86 × 10^−12^	1.52 × 10^5^	1000
Mean value ± σ_es_ *	1.02 × 10^−11^ ± 0.07 × 10^−11^	2.04 × 10^4^ ± 0.70 × 10^4^	1000	1.01 × 10^−11^ ± 0.03 × 10^−11^	1.12 × 10^5^ ± 0.39 × 10^5^	1000

(q_0_ = 0, in all cases); * σ_es_: standard deviation based on the entire population given as arguments (MC Office Excel function STDEV.P).

**Table 4 materials-13-01915-t004:** The evaluation of kinetic dependences by six different models (Table 1) based on the values of WSOS/DF (weighted sum of squares divided by degrees of freedom) characterizing the agreement between the experimental (22 ± 1 °C) and calculated data.

Sorbent	Experiment No.	WSOS/DF					
		DM	FD	ID	RLD	CR	GD
*n*HAp	1	59.70	1860	5.35	**5.47**	1320	40.30
	2	8.28	186	7.45	**7.25**	92.60	10.10
	3	8.06	461	2.60	**2.71**	250	27.60
	Mean value ± σ_es_ *	25.34 ± 24.29	835.67 ± 732.96	5.13 ± 1.99	**5.14 ± 1.87**	554.20 ± 545.30	26.00 ± 12.38
*n*TiO_2_	1	48.70	54.70	6.72	**7.63**	52.80	9.99
	2	5.76	757	1.23	**3.18**	36.30	9.42
	3	33.40	258	8.09	**16.10**	239	5.43
	Mean value ± σ_es_ *	29.29 ± 17.77	356.57 ± 295.06	5.35 ± 2.97	**8.96 ± 5.36**	109.37 ± 91.91	8.27 ± 2.04

* σ_es_: standard deviation based on the entire population given as arguments (MC Office Excel function STDEV.P).

**Table 5 materials-13-01915-t005:** The values of over-all kinetic, *K_RLD_*, and diffusion, *D*, coefficients, and half-life of sorption, *t*_1/2_, for the diffusion in reacted layer (RLD) model.

Sorbent	Experiment No.:	*K*_RLD_ ± σ (cm^3^·g^−1^·min^−1^)	Mean * *K*_RLD_ ± σ_es_ (cm^3^·g^−1^·min^−1^)	Mean * *D* ± σ_es_ (cm^2^·min^−1^)	Mean * *t*_1/2_ ± σ_es_ (min)
*n*HAp	1	1.24 × 10^−1^	5.03 × 10^−1^ ± 2.72 × 10^−1^	2.50 × 10^−12^ ± 1.80 × 10^−12^	0.75 ± 0.18
	2	7.53 × 10^−1^			
	3	6.31 × 10^−1^			
*n*TiO_2_	1	2.82 × 10^−2^	4.01 × 10^−2^ ± 2.44 × 10^−2^	1.60 × 10^−14^ ± 0.96 × 10^−14^	0.51 ± 0.32
	2	1.80 × 10^−2^			
	3	0.74 × 10^−2^			

* σ_es_: standard deviation based on the entire population given as arguments (MC Office Excel function STDEV.P).

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
