# Peer review of "Determination, Modeling and Evaluation of Kinetics of 223Ra Sorption on Hydroxyapatite and Titanium Dioxide Nanoparticles"

_materials, 2020, doi:10.3390/ma13081915_

Round 1

Reviewer 1 Report

Page 1 Line 22: In the text Abstract it is necessary to correct the notation of concentration.

Page 2 line 27: Sorption capacity for HAp denoted as "0.5 g of Pb can be removed by 1 g HAp". And line 60 at the same page the sorption capacity of TiO2 denoted as “5 mg/g”. The sorption capacity should be indicated, if you are talking about it, in the same quantities (mg/g).

In the part Materials and Methods, the concentration range is not specified. There are no methods characterizing sorbent powders. Further, table 2 indicates the equivalent diameter, surface edge sites, surface layer sites, specific mass. What methods determined these parameters?

Please, explain the selection of six kinetic models. Why were models of pseudo first and pseudo second orders and intraparticle diffusion not considered?

Please indicate the initial concentrations of radium-223 in the caption to Figures 1.

Pages 5-6 lines 178-184: It is difficult to follow the author's idea. It would be better Tables 3 and 4 should be interchanged so that the reader can more clearly understand the choice of only two kinetic models. More detailed explanations should also be given.

Page 7 line 206: give more detailed explanations of how bulk density and mass density affect the diffusion coefficient.

Have the authors made estimates of the sorption capacity of the studied sorbents concerning radium-223? And can this be compared with other sorbents or other ions?

The conclusions presented are not in line with the objectives of the work. There are no main conclusions about the role of surface characteristics of sorbents on the kinetics of the sorption of a radionuclide.

Author Response

Answers to the Reviewer 1

Firstly thank you for your suggestions and comments that certainly improved the quality of our paper. Please see the answers bellow and the revised manuscript. 

Page 1 Line 22: In the text Abstract it is necessary to correct the notation of concentration.

The concentration range was correct, but superscript was missing – now added.

Page 2 line 27: Sorption capacity for HAp denoted as "0.5 g of Pb can be removed by 1 g HAp". And line 60 at the same page the sorption capacity of TiO2 denoted as “5 mg/g”. The sorption capacity should be indicated, if you are talking about it, in the same quantities (mg/g).

The sorption capacity for HAp was changed - Page 2 line 47: It was found that sorption capacity of HAp was up to 500 mg/g for Pb.

In the part Materials and Methods, the concentration range is not specified.

Initial concentrations are shown in Table 3 in the part Results and discussion. There is only the equation for concentration calculation. In the part Materials and methods the results of our calculations are not given due to their summarisation in the next part.

There are no methods characterizing sorbent powders.

This paper is not focused on preparation and characterization of studied materials. These data were already published in a separate paper - citation 6 (Kukleva et al. 2019). Page 3 line 114: Detailed description of the materials preparation and characterization was published by Kukleva et al. [6] earlier.

Further, table 2 indicates the equivalent diameter, surface edge sites, surface layer sites, specific mass. What methods determined these parameters?

These values and determination methods are also clearly described earlier and the citations are specified in Table 2 description - citations 6 and 10 - Kukleva et al. 2019, Suchankova et al. 2020).

Please, explain the selection of six kinetic models. Why were models of pseudo first and pseudo second orders and intraparticle diffusion not considered?

We have had good experience with these models for many years, namely to describe the kinetics of sorption (and desorption) taking place in heterogeneous systems. They were derived for different types of diffusion control processes (see DM, FD, ID, RLD and GD models) and also for the combination of diffusion and chemical reaction (see CR model). We emphasize applicability for heterogeneous systems. We know the Intrapartical model, but we believe that the models we use, especially ID, RLD and GD, are analogous to it and, moreover, differentiate different types of sorbents. The DM model, or two-film model, was derived specifically for liquid-liquid extraction systems. We are also familiar with the use of the pseudo-first and pseudo-second order kinetic models, but from a physical standpoint, for heterogeneous systems, it is a regression relationship rather than a equation for the description of the processes taking place in solid-liquid systems.

Please indicate the initial concentrations of radium-223 in the caption to Figures 1.

Initial concentrations were added to Figures 1.

Pages 5-6 lines 178-184: It is difficult to follow the author's idea. It would be better Tables 3 and 4 should be interchanged so that the reader can more clearly understand the choice of only two kinetic models. More detailed explanations should also be given.

We do not think that switching tables would be useful, since it is necessary to do the calculation first by inserting the input data. The resulting WSOS / DF values are than obtained and the suitability of a particular model is assessed. However, as the text suggests, not only the WSOS / DF criterion value is taken into account, but also the physico-chemical properties of the sorbent -- here we choose the RLD model because the diffusion of the sorbed component, i.e. 223Ra, takes place in the direction away from the surface of the sorbent particle, it is gradually sorbed, i.e., 223Ra diffuses through the already reacted layer. However, the paragraph describing both tables was divided.

Page 7 line 206: give more detailed explanations of how bulk density and mass density affect the diffusion coefficient.

Although the direct dependence of the diffusion coefficient on these parameters is unknown, however, for example, in the case of RN migration in a compressed bentonite layer, the so-called engineering barrier, the diffusion coefficient decreases to some extent with the degree of compression. By increasing the compression, the volume of a given layer decreases, i.e. the bulk density increases (this is a function of the mass density) and the porosity changes. However, we admit that the diffusion coefficient considered here cannot be evaluated in this way - - therefore, we leave in the text only its dependence on the temperature and composition of the liquid phase, more precisely on the form of the existence of radium in solution, i.e., on  its size (influenced, for example, by the hydration of the ion in question).

Have the authors made estimates of the sorption capacity of the studied sorbents concerning radium-223? And can this be compared with other sorbents or other ions?

The sorption capacity for radium under the conditions of our experiments can be calculated using the quantity Kd and the input concentration C0: so, for TiO2 ... 1.13e-6, and for HAp ... 2.08e-7 mol / kg. Sorption experiments were performed earlier as a dependence of pH, but the sorption capacity was not estimated. The dependance on pH  was compared with other sorbent - SPIONs. Citation Suchankova et al. 2020 [10]

The conclusions presented are not in line with the objectives of the work. There are no main conclusions about the role of surface characteristics of sorbents on the kinetics of the sorption of a radionuclide.

The conclusions part is now structuated according to the objectives of our work in the introduction.

Reviewer 2 Report

Materials-771599

Title: Determination, modeling and evaluation of kinetics of 223Ra sorption on hydroxyapatite and titanium dioxide nanoparticles

Review:

This manuscript investigated the sorption kinetics of radium on hydroxyapatite and titanium dioxide nanoparticles. Six kinetic models were applied to analyze the experimental data and the diffusion model in a reacted layer was selected as the best-fit model. This study is a part of a series of work from the authors’ previous work [ref. 6, 10] and focusing on the kinetics. In order to validate the kinetic model selected in the present study, an additional explanation is preferable. I would suggest publication of the manuscript in the journal, after comments below are addressed.

  • Page 3, line 116

Please give an information on the ionic strength of the sample solutions.

  • Page 4, line 149

In the results and discussion section, all six models were applied and RLD models was selected for further discussion on the kinetic analysis, although it is written here that the ID-model was used to demonstrate the kinetics. Please check the sentence.

  • Page 5, line 170

Please describe how to determine the crystalline size and equivalent diameter of NPs. Please give errors of the crystalline size of nHAp and nTiO2 in Table 2.

  • Page 6, line 182

It is not clear the reason to conclude that the RLD model was the best model to explain the experimental results. The WSOS/Df values in Table 4 are found to be lower for the ID model than those for RLD model. In Fig. 1(b), the calculated curve dose not seem to well reproduce the experimental values.

Additional explanation is needed to select the RLD model.

  • Page 6, line 189

Please give errors of WSOS/DF values to compare the values in different models.

  • Page 6, line 195

“all three repeated experiments” seems clearer for readers.

  • Page 6, line 202

In the analysis, dose the deviation of particle size included to determine the parameters in Table 3?

  • Page 7, line 227

This paragraph seems not clear. Please clarify the authors’ claim.

Author Response

Answers to the Reviewer 2

Firstly thank you for your suggestions and comments that certainly improved the quality of our paper. Please see the answers bellow and the revised manuscript. 

  • Page 3, line 116

Please give an information on the ionic strength of the sample solutions.

It is not possible to give correct information on ionic strength of sample solutions due to the buffers. Buffers were prepared by addition of 0.2 M NaOH solution into the mixture of acids until desired pH was reached. The amount of NaOH added is unknown. However it influence ionic strength slightly and it could be declared that for pH 6 ionic strength was in the range from 0.08 to 0.09 M, and for buffer with pH 8 in the range from 0.16 to 0.17 M. Ion strength was not adjusted. Due to the fact, that this data are not verified and were not included in calculations their presence in the article is not necessary on our opinion. In general, the calculated data do not correspond with data available in literature, so they are unreliable. [see e.g.: Mongay C., Cedra V. Annali di Chimica, 64, 1947, p. 409-412]

  • Page 4, line 149

In the results and discussion section, all six models were applied and RLD models was selected for further discussion on the kinetic analysis, although it is written here that the ID-model was used to demonstrate the kinetics. Please check the sentence.

It was a mistake and it was changed.

  • Page 5, line 170

Please describe how to determine the crystalline size and equivalent diameter of NPs. Please give errors of the crystalline size of nHAp and nTiO2 in Table 2.

These data are described and summarized in earlier publication Kukleva et al. 2019 [6]. 

  • Page 6, line 182

It is not clear the reason to conclude that the RLD model was the best model to explain the experimental results. The WSOS/Df values in Table 4 are found to be lower for the ID model than those for RLD model. In Fig. 1(b), the calculated curve dose not seem to well reproduce the experimental values.

Additional explanation is needed to select the RLD model.

These data are described and summarized in earlier publication Kukleva et al. 2019 [6]. These data are described and summarized in earlier publication Kukleva et al. 2019 [6]. We have had good experience with these models for many years, namely to describe the kinetics of sorption (and desorption) taking place in heterogeneous systems. They were derived for different types of diffusion control processes (see DM, FD, ID, RLD and GD models) and also for the combination of diffusion and chemical reaction (see CR model). We emphasize applicability for heterogeneous systems. We know the Intrapartical model, but we believe that the models we use, especially ID, RLD and GD, are analogous to it and, moreover, differentiate different types of sorbents. The DM model, or two-film model, was derived specifically for liquid-liquid extraction systems. We are also familiar with the use of the pseudo-first and pseudo-second order kinetic models, but from a physical standpoint, for heterogeneous systems, it is a regression relationship rather than a equation for the description of the processes taking place in solid-liquid systems. 

The resulting WSOS / DF values are than obtained and the suitability of a particular model is assessed. However, as the text suggests, not only the WSOS / DF criterion value is taken into account, but also the physico-chemical properties of the sorbent -- here we choose the RLD model because the diffusion of the sorbed component, i.e. 223Ra, takes place in the direction away from the surface of the sorbent particle, it is gradually sorbed, i.e., 223Ra diffuses through the already reacted layer.

  • Page 6, line 189

Please give errors of WSOS/DF values to compare the values in different models.

They were added into Table 4.

  • Page 6, line 195

“all three repeated experiments” seems clearer for readers.

 It was added.

  • Page 6, line 202

In the analysis, dose the deviation of particle size included to determine the parameters in Table 3?

For calculations it is not necessary, only mass is used for calculations.

  • Page 7, line 227

This paragraph seems not clear. Please clarify the authors’ claim.

This paragraph was mistakenly used template. It was deleted.

Round 2

Reviewer 1 Report

After corrections in the manuscript, I believe it is now in good standing to be published.

Reviewer 2 Report

The has been modified according to the comments. I suggest the publication of the manuscript in the present form.